# Functionally Isolated Sarcoplasmic Reticulum in Cardiomyocytes: Experimental and Mathematical Models

**DOI:** 10.3390/bioengineering12060627

**Published:** 2025-06-09

**Authors:** Diogo C. Soriano, Rosana A. Bassani, José W. M. Bassani

**Affiliations:** 1Department of Electronics and Biomedical Engineering, School of Electrical and Computer Engineering, Universidade Estadual de Campinas (UNICAMP), Av. Albert Einstein 400, Campinas 13083-852, SP, Brazil; bassani@unicamp.br; 2Center for Engineering, Modeling and Social Sciences, Universidade Federal do ABC (UFABC), Alameda da Universidade, s/n, São Bernardo do Campo 09606-045, SP, Brazil; 3Brazilian Laboratory for Cellular Calcium Research (LabNECC), Center for Biomedical Engineering, Universidade Estadual de Campinas (UNICAMP), R. Alexander Fleming 163, Campinas 13083-881, SP, Brazil; rbassani@unicamp.br

**Keywords:** sarcoplasmic reticulum, Ca^2+^ transport, mathematical modeling, cardiomyocytes, β-adrenergic stimulation, 2,5-di-tert-butylhydroquinone, caffeine

## Abstract

The interaction among the various Ca^2+^ transporters complicates the assessment of isolated systems in an intact cell. This article proposes the functionally isolated SR model (FISRM), a hybrid (experimental and mathematical) approach to study Ca^2+^ cycling between the cytosol and the sarcoplasmic reticulum (SR), the main source of Ca^2+^ for contraction in mammalian cardiomyocytes. In FISRM, the main transmembrane Ca^2+^ transport pathways are eliminated by using a Na^+^, Ca^2+^-free extracellular medium, and SR Ca^2+^ release is elicited by a train of brief caffeine pulses. Two compounds that exert opposite effects on the SR Ca^2+^ uptake were characterized by this approach in isolated rat ventricular cardiomyocytes. The experimental FISRM was simulated with a simple mathematical model of Ca^2+^ fluxes across the SR membrane, based on a previous model adapted to the present conditions. To a fair extent, the theoretical model could reproduce the experimental results, and confirm the main assumption of the experimental model: that the only relevant Ca^2+^ fluxes occur across the SR membrane. Thus, the FISRM seems to be a valuable framework to investigate the SR Ca^2+^ transport in intact cardiomyocytes under physiological and pathophysiological conditions, and to test therapeutic approaches targeting SR proteins.

## 1. Introduction

The Ca^2+^-induced Ca^2+^ release mechanism plays a central role in excitation–contraction coupling (ECC) in the mammalian myocardium. Briefly, sarcolemmal depolarization during the action potential allows Ca^2+^ influx via sarcolemmal voltage-dependent L-type Ca^2+^ channels, which causes the release of a much larger amount of Ca^2+^ from the intracellular store in the sarcoplasmic reticulum (SR). The resultant increase in the cytosolic free Ca^2+^ concentration ([Ca^2+^]i) promotes interaction of the ion with troponin C, present in the contractile apparatus, leading to the development of contraction, and ultimately to blood pumping to the circulation. SR Ca^2+^ release is triggered by interaction of Ca^2+^ with Ca^2+^ release channels in the SR membrane, also called ryanodine receptors (RyR) [1,2,3].

Cell relaxation, fundamental for ventricular refilling with blood to be ejected in the next contraction, is due to Ca^2+^ dissociation from troponin C resulting from removal of cytosolic Ca^2+^ by transporters stimulated by the elevated [Ca^2+^]i. This is accomplished mainly by the SR Ca^2+^-ATPase (SERCA), which pumps Ca^2+^ back into the SR and replenishes its stores, and, to a lesser degree, by the sarcolemmal Na^+^-Ca^2+^ exchanger (NCX), which mediates most of Ca^2+^ efflux from the cell. Other transporters, namely the plasma membrane Ca^2+^-ATPase (PMCA) and the mitochondrial Ca^2+^ uniporter (MCU), with low transport capacity and affinity for Ca^2+^, respectively, seem to play a minor role in this process (<2% [4,5]).

These mechanisms define an integrated process, in which intracellular and transsarcolemmal Ca^2+^ transport modulate the ECC effectiveness. The balance between Ca^2+^ influx and efflux determines the amount of Ca^2+^ available for uptake by SERCA, and, therefore, the SR Ca^2+^ content. The latter determines the fraction of this content to be released at the next twitch [6,7,8], which impacts not only contraction amplitude, but also the NCX-mediated Ca^2+^ removal. NCX function, in turn, may affect transmembrane electric potential, action potential configuration and SR Ca^2+^ content, evidencing an important feedback control mechanism that regulates the inotropic function in different physiological and pathophysiological scenarios [6,9]. On the other hand, Ca^2+^-dependent regulation of Ca^2+^ transporters, by, e.g., Ca^2+^-calmodulin-dependent enzymes, introduces additional complexity levels to an already complex interplay (for a review, see, e.g., [10,11]).

Understanding ECC and the SR function is important to identify focuses for the therapy of cardiac diseases. The SR is the main Ca^2+^ contributor for contraction in both healthy and failing mammalian myocardium. The SR-cytosol Ca^2+^ cycling is estimated to account for more than 70% of the Ca^2+^ fluxes during an activity cycle in mammalian cardiomyocytes, reaching 90% in the adult rat ventricle [2,4,5,12]. Accordingly, several studies have shown that defects in SR Ca^2+^ cycling are associated with impaired systolic/diastolic function and arrhythmias, and that SR proteins seem to be promising targets for heart failure therapy (e.g., [9,13,14,15]. However, finding an experimental model suitable for the study of the myocardial SR function is not simple. While it is difficult to reproduce physiological intracellular conditions when using isolated SR vesicles and permeabilized cardiomyocytes, the variety of Ca^2+^ flux pathways in intact isolated cells makes it difficult to isolate those attributable to SR Ca^2+^ release and uptake.

In this report, we propose an experimental model and its theoretical framework, the functionally isolated SR model (FISRM), for studying SR-cytosol Ca^2+^ cycling in intact isolated myocytes under electrical rest. Ca^2+^ transport via NCX, the main SERCA competitor, is thermodynamically suppressed by removing Na^+^ and Ca^2+^ from the extracellular medium [4,5], which also renders the cell unable to develop action potentials and abolishes Ca^2+^ influx. The application of brief caffeine pulses is used to trigger Ca^2+^ transients due solely to SR Ca^2+^ release, whereas [Ca^2+^]i decline is majorly accomplished by sequestration by SERCA. This system was paralleled by a mathematical model in which the phenomena underlying SR Ca^2+^ release by caffeine were introduced as an adaptation of a realistic mathematical model of rodent cardiomyocyte [16]. The results indicate that the model is suitable for investigation of the SR-cytosol Ca^2+^ transport in intact cells without significant interference of other transporters.

## 2. Materials and Methods

### 2.1. Cell Isolation and Perfusion Solutions

All procedures for care and use of the animals (the same as described by Penna and Bassani [17]) were in agreement with Brazilian laws and were approved by the institutional Committee for Ethics in Animal Use (CEUA/UNICAMP, N. 775-1 and 952-1).

Myocytes were isolated from the left ventricle of male adult Wistar rats by coronary perfusion with collagenase I, followed by mechanical dispersion [17]. After isolation, cell suspensions in cardioplegic solution (composition in mM: 30 KCl; 70 glutamic acid; 10 KH_2_PO_4_; 5 HEPES; 1 MgCl_2_; 11 glucose; 20 taurine; pH 7.4) were kept at 4 °C until use. Experiments were performed at 24–25 °C. Cells were perfused with modified Tyrode’s solution (TyN), with the following composition (mM): 140 NaCl, 6 KCl, 1.5 MgCl_2_, 5 HEPES, 11 glucose and 1 CaCl_2_; pH 7.4 at 24 °C. In the Na^+^-, Ca^2+^-free solution (Ty00), LiCl and EGTA replaced for NaCl and CaCl_2_ in an equimolar fashion. Caffeine (10 mM) was dissolved in Ty00 (Caf00 solution).

### 2.2. Cell Contraction and [Ca^2+^]i Measurement

Myocytes were plated in a perfusion chamber of which the bottom was a collagen-treated coverslip. Cells were perfused with TyN and electrically stimulated via a pair of platinum electrodes (10 ms-long biphasic, rectangular voltage pulses delivered at 0.5 Hz). Unloaded cell shortening and Ca^2+^ transients were recorded simultaneously.

Systolic cell shortening (cell contraction amplitude, ΔL) was recorded with a video-edge detector and expressed as percent of resting cell length, which was measured with the aid of objective micrometer [17].

For [Ca^2+^]i measurement, cells were incubated with the cell-permeant Ca^2+^ indicator fluo-3 AM (1 µM; Life Technologies, Grand Island, NY, USA) for 20 min, followed by 20 min perfusion with TyN for fluo-3 deesterification. The dye was excited at 480 ± 20 nm, and its emission was collected at 515–560 nm using a fluorescence microscopy system previously described [18]. The background-subtracted emitted signal was converted to [Ca^2+^]i according to Shannon et al. [19]:(1)$$[{Ca}^{2+}]i=\frac{K_{d}\cdot \frac{F}{F_{0}}}{\frac{K_{d}}{[Ca^{2+}{]}_{d}}-\frac{F}{F_{0}}+1}$$
where F/F_0_ is the fluorescence emission normalized to that at diastole; K_d_ is the Ca^2+^-fluo-3 apparent Ca^2+^ dissociation constant (1.1 μM [20]), and [Ca^2+^]_d_ is the diastolic [Ca^2+^]i taken as 0.23 μM from calibrated measurements with indo-1 [7,12].

### 2.3. The Experimental Protocol

The main inlets of the perfusion chamber admitted TyN or Ty00 solution, which bathed all cells in the chamber (~3 mL·min^−1^). A single cell was selected, and a pair of glass micropipettes was positioned aiming at it, so that solution delivered through them would perfuse the chosen cell and immediate surroundings. Each pipette delivered either Ty00 or Caf00 solution. Solutions were switched on and off with solenoid valves (LFAA mod. 12016-18H, The Lee Company, Westbrook, CT, USA) controlled by an oscillator circuit placed close to the micropipettes and chamber inlets.

Caffeine, a well-known RyR opener [21], was used to evoke SR Ca^2+^ release. Application of short (60 ms-long) caffeine pulses was implemented by rapid switching of the perfusion solution from the Tyr00 pipette to the Caf00 pipette (delay < 7 ms). Before and during caffeine application, the whole chamber was perfused with Ty00 via one of its main inlets.

Initially, cells were electrically stimulated for 2–5 min under perfusion with TyN for stabilization of Ca^2+^ transients and of free and bound [Ca^2+^] within the SR. Then, electrical stimulation was interrupted, and perfusion was switched to Ty00 through both one chamber inlet and one of the micropipettes. The removal of extracellular Na^+^ and Ca^2+^ ions abolished both Ca^2+^ efflux via NCX and Ca^2+^ influx, allowing isolation of intracellular SR-cytosol Ca^2+^ fluxes [4]. After 30 s perfusion with Ty00, caffeine pulses were applied by switching perfusion through the micropipette from Ty00 to Caf00, and then back to Ty00 60 ms later. Successive caffeine pulses (5–8) were applied at 7 s intervals to allow complete caffeine washout between pulses.

In some experiments, SR Ca^2+^ content was estimated to check the possibility of significant loss of SR Ca^2+^ during the caffeine pulses due to removal of the released Ca^2+^ by PMCA and MCU [4,5]. The SR Ca^2+^ load was estimated as in Bassani and Bers [22]. Briefly, it was defined as the total amount of Ca^2+^ in the SR released by caffeine ([Ca^2+^]_SR_, i.e., the sum of free and bound [Ca^2+^]), and expressed as µmoles Ca^2+^ per liter of non-mitochondrial cell water. The cytosolic total [Ca^2+^] ([Ca^2+^]_T_) was calculated from the recorded [Ca^2+^]i during sustained perfusion with Caff00 as follows:(2)$${\left[{Ca}^{2+}\right]}_{T}=\left[{Ca}^{2+}\right]i+\left(\frac{B_{fmax}\cdot \left[{Ca}^{2+}\right]i}{\left[{Ca}^{2+}\right]i+K_{df}}\right)+\left(\frac{B_{emax}\cdot \left[{Ca}^{2+}\right]i}{\left[{Ca}^{2+}\right]i+K_{de}}\right)$$
where B_fmax_ is the assumed maximum concentration of Ca^2+^ binding sites in fluo-3 (4 μM) and K_df_ is the Ca^2+^-fluo-3 apparent Ca^2+^ dissociation constant (1.1 μM [20]); B_emax_ and K_de_ are, respectively, the maximum concentration of endogenous passive Ca^2+^ binding sites (300 μM) and the respective apparent dissociation constant (0.52 μM) determined in permeabilized rat ventricular myocytes [23]. [Ca^2+^]_SR_ was considered as the difference between the [Ca^2+^]_T_ values at the Ca^2+^ transient peak and immediately before the caffeine application.

### 2.4. SERCA Stimulation and Inhibition

Two pharmacological treatments aiming at SERCA were used to test FISRM. For SERCA stimulation, cells were exposed to 10 nM isoproterenol (ISO), a β-adrenoceptor agonist that promotes intracellular accumulation of the second messenger 3′-5′-cyclic adenosine monophosphate (cAMP), which activates the cAMP-dependent protein kinase (PKA). This enzyme phosphorylates several proteins, including phospholamban (PLN), an endogenous SERCA inhibitor. Phosphorylation relieves PLN allosteric inhibition of SERCA, increasing the enzyme affinity for Ca^2+^, and thus the rate of SR Ca^2+^ uptake [15]. To cause partial inhibition of SERCA activity, 2,5-di-(tert-butyl)-1,4-benzohydroquinone (tBQ, 5 µM) was added to the perfusion solution 10 min before and during Ca^2+^ transient recording. This compound inhibits SERCA in muscle and non-muscle cells in a reversible fashion [24,25].

β-adrenoceptor activation with ISO has a well-know stimulating effect on sarcolemmal L-type Ca^2+^ channels [26,27], whereas tBQ was reported to inhibit the L-type Ca^2+^ current (I_CaL_) in cardiac myocytes [25]. Although these effects might complicate the analysis of SR-cytosol Ca^2+^ cycling when SR Ca^2+^ release depends on I_CaL_, they should not be a problem in the FISRM, in which this current is absent.

### 2.5. The Mathematical Model

A numerical analysis describing the FISRM framework can be introduced to the model proposed by Bondarenko et al. [16], which reproduces many aspects of cardiomyocyte electrophysiology and Ca^2+^ dynamics. Briefly, the main intracellular Ca^2+^ fluxes modelled are: (a) Ca^2+^ uptake from the cytosol into the network SR (J_up_); (b) Ca^2+^ release from junctional SR (J_rel_), in addition to SR Ca^2+^ leak (J_leak_); and (c) Ca^2+^ transfer between network and junctional SR (J_tr_) and between the subsarcolemmal space and the bulk myoplasm (J_xfer_). Ca^2+^ buffering by troponin C and calmodulin (cytosol) and by calsequestrin (intra-SR) is also considered. Mathematically, J_up_ and J_rel_, the most relevant fluxes in FISRM, are given by:(3)$$J_{up}=\frac{V_{max}\cdot [Ca^{2+}{]}_{i}^{2}}{K_{m,up}+[Ca^{2+}{]}_{i}^{2}}$$(4)$$J_{rel}=v\cdot (P_{O1}+P_{O2})\cdot ([Ca^{2+}{]}_{JSR}-[Ca^{2+}{]}_{SS})\cdot P_{RyR}$$
where V_max_ and K_m,up_ are, respectively, the maximum velocity and the Michaelis–Menten constant for Ca^2+^ uptake by SERCA; v is the maximum total RyR permeability; P_O1_ and P_O2_ are the RyR fractions in the O1 and O2 (open) states; [Ca^2+^]_JSR_ and [Ca^2+^]_SS_ are the [Ca^2+^] in the junctional SR and subsarcolemmal space, respectively; and P_RyR_ is the RyR modulation factor for SR Ca^2+^ release.

The state equations that rule RyR behavior in this model are based on the proposal of Keizer and Levine [28] modified by Jafri et al. [29]. In short, two different channel open and closed states are considered, with transition rates ($k_{x}^{+}$ and $k_{x}^{-}$) that may or may not depend on [Ca^2+^]_SS_, as illustrated in the schematic transition diagram for RyR (Figure 1). RyR Ca^2+^ binding sites that regulate channel opening act as [Ca^2+^]_SS_ sensors.

In the present simulation, caffeine was assumed to diffuse passively across the sarcolemma with a first order dynamics and to modulate $k_{a}^{+}$ and $k_{c}^{-}$ rates, favoring RyR transition to the open states (Figure 1). In this case, the temporal evolution of the modulation factor G_caff_(t) can be described by:(5)$${\dot{G}}_{caff}(t)=\frac{{G_{c}}_{aff}(\infty )-G_{caff}(t)}{\tau_{caff}}$$
where $G_{caff}(\infty )$ is defined as 1 and 150 in the absence and presence of caffeine, respectively (values chosen empirically); τ_caff_ (time constant of increase in intracellular caffeine concentration) was considered as 50 ms, based on experimental measurements [30]. The $k_{a}^{+}$ and $k_{c}^{-}$ rates are then redefined as:(6)$$k_{a}^{+}\prime =G_{caff}(t)\cdot k_{a}^{+}$$(7)$${k_{c}^{-}}^{\prime }=G_{caff}\left(t\right)\cdot k_{c}^{-}$$

P_RYR_, the RyR modulation factor proposed by Bondarenko et al. [16], was adjusted for better simulation of the experimental Ca^2+^ transients triggered by caffeine. Here, P_RYR_(t) also assumed a first order dynamic behavior given by:(8)$${\dot{P}}_{RYR}(t)=\frac{P_{RYR}(\infty )-P_{RYR}(t)}{\tau_{RYR}}$$

In this case, by setting $P_{RYR}\left(\infty \right)=0$ and $\tau_{RYR}=75$ ms in the absence of caffeine, and $P_{RYR}\left(\infty \right)=0.25$ and $\tau_{RYR}=10\mathrm{ms}$ during the first 20 ms of the caffeine pulse, a bell-shaped curve for $P_{RYR}\left(t\right)$ with duration similar to of the whole caffeine pulse is obtained, in analogy to that obtained by Bondarenko et al. [16] under electrical pacing. During exposure to ISO and tBQ, $P_{RYR}\left(\infty \right)$ was adjusted to 0.15 and 0.40, respectively. For more details, see the Appendix A, in which a schematic representation of the mathematical FISRM is shown (Appendix A).

### 2.6. Data and Statistical Analysys

The analyzed variables of twitch contractions/transients (evoked by electrical stimulation) and of caffeine evoked contractions/transients were: (a) the mean amplitude of contractions (ΔL) and of Ca^2+^ transients (Δ[Ca^2+^]i, i.e., the difference between the peak and previous diastolic [Ca^2+^]i values); (b) the half-time (t_0.5_) values of mechanical relaxation and [Ca^2+^]i decline, measured in 4–5 successive similar waveforms; and (c) the SR Ca^2+^ content. Data are presented as the means and respective standard deviations. The variables were compared with Student’s *t*-test for paired samples, and the statistical significance was set as *p* < 0.05.

## 3. Results

### 3.1. Caffeine Pulses

Figure 2 shows typical Ca^2+^ transients evoked by a train of brief caffeine pulses applied in the absence of extracellular Na^+^ and Ca^2+^, thus defining the FISR experimental paradigm of Ca^2+^ cycling. The fast rise in [Ca^2+^]i indicates that caffeine is able to quickly cross the sarcolemma and induce Ca^2+^ release from the SR. The full recovery of interpulse diastolic [Ca^2+^]i levels indicates that, under these experimental conditions, caffeine quickly diffuses out of the cell during its washout. A gradual decrease in the transient peak was often observed after a few pulses, and, for this reason, further measurements and comparisons were carried out only up to the 4–5th pulse.

A particularly important assumption of the FISR approach is the maintenance of stable SR Ca^2+^ load over successive caffeine pulses. Figure 3 illustrates the experiment performed to test this premise. Initially, myocytes were electrically stimulated until stabilization of transients. Perfusion was then switched to Ty00 and, ~45 s later, sustained perfusion with Caf00 was applied to evoke complete release of the Ca^2+^ stored in the SR, so that the control SR Ca^2+^ content could be estimated (panel A). The experiment was repeated, but five brief caffeine pulses were applied before sustained Caf00 application (panel B). In a set of seven independent experiments, the amount of Ca^2+^ stored in the SR after caffeine pulses (83.2 ± 17.6 µM) was not significantly different for the control value (86.6 ± 13.1 µM; *p* = 0.55). This confirms that application of five repeated caffeine pulses does not result in significant SR Ca^2+^ loss, thus allowing characterizing the model, under these conditions, as a conservative system.

The FISRM mathematical formulation was able to reproduce the behavior of the experimental model, as shown in Figure 4. In the presence of Ty00 solution (in which [Na^+^]o and [Ca^2+^]o were decreased from 140 and 1.8 mM, respectively, to ~1 µM) and absence of electrical activity, caffeine pulses were able to evoke Ca^2+^ transients a little larger than those evoked by electrical stimulation in both models, possibly due to greater RyR activation by the compound than by I_CaL_ and/or lack of NCX-mediated Ca^2+^ efflux. A small, but progressive decrease in the amplitude of the transients was observed, as seen in the experimental model.

### 3.2. Effects of β-Adrenergic Stimulation by ISO

SERCA activity is a key target of the β-adrenoceptor-cAMP-PKA signaling cascade in the generation of positive inotropic (increase in contraction amplitude) and lusitropic (acceleration of relaxation) effects in the myocardium. As can be seen in Table 1 and Figure 5, an ISO concentration as low as 10 nM could nearly double the twitch Δ[Ca^2+^]i and ΔL, and decrease by ~25% the t_0.5_ of [Ca^2+^]i decline and relaxation. ISO also produced a small, but significant increase in SR Ca^2+^ load. These results are consistent with previous observations from this lab (e.g., [17,18] and with the well-known increase in I_CaL_ and in SERCA activity by stimulation of myocardial β-adrenoceptors (e.g., [27,31]).

Nevertheless, the amplitude of caffeine-triggered Ca^2+^ transients was not changed by ISO, although this agent could produce a significant positive lusitropic effect, reducing the t_0.5_ of [Ca^2+^]i decline by a similar extent as in the twitch (~25%; Table 2 and Figure 6).

The action of ISO was simulated in the mathematical model as a 40% increase in the permeability of the sarcolemmal Ca^2+^ L-type channels (P_CaL_), associated with a 35% decrease in the SERCA K_m_ [8], leaving unaltered the RyR parameters. The latter change simulated the relief of SERCA inhibition due to phopholamban phosphorylation. As shown in Figure 7, the mathematical FISRM reproduced the marked increase observed experimentally in Δ[Ca^2+^]i of twitches, but not of caffeine-evoked transients. [Ca^2+^]i decline in twitches and caffeine transients was also faster, although in the latter it was accelerated in the later phase of the decay, rather than at the t_0.5_ level, as in the experiments. Nevertheless, in both experiment and simulation, a 25–30% decrease in the total duration of [Ca^2+^]i decline was observed.

### 3.3. Effects of SERCA Inhibition by tBQ

The effects of partial SERCA inhibition with 5 µM tBQ were also examined in paced cells and in the FISRM. The compound exerted effects opposite to those of ISO: a negative inotropic effect, which was rather variable in magnitude among cells, resulting marginally or not statistically significant. But a stronger negative lusitropic effect was observed for both contraction and Ca^2+^ transients, in addition to a significant loss of the SR Ca^2+^ load (Table 3). Electrically evoked twitch Ca^2+^ transients and contractions recorded before and during exposure to tBQ are shown in Figure 8. In the experimental FISRM, tBQ depressed the amplitude of caffeine-triggered Ca^2+^ transients, and, as observed for twitches, prolonged [Ca^2+^]i decline (Table 2; Figure 9).

The action of tBQ was simulated in the mathematical model as a 40% decrease in the maximal velocity of Ca^2+^ uptake by SERCA (V_max_). This change decreased Δ[Ca^2+^]i by ~40% and doubled the t_0.5_ for [Ca^2+^]i decay of twitches, effects that were qualitatively similar to those seen experimentally, although of greater magnitude. Amplitude decrease and lengthening were also observed for simulated caffeine-evoked transients, but changes were less pronounced than those in twitches (Figure 10). Additionally, prolongation of [Ca^2+^]i decay at 50% decline was smaller than that seen experimentally, but comparable at 90% decay.

## 4. Discussion and Conclusions

The present study proposes the FISRM, devised to study the bidirectional Ca^2+^ transport between the cytosol and the SR in intact, contracting, isolated myocytes. The condition of SR functional isolation can be easily reversed by simply switching perfusion to a physiological solution containing Na^+^ and Ca^2+^, which restores the cell ability to develop electrical activity and the transsarcolemmal Ca^2+^ transport. In this model, SR Ca^2+^ release is achieved by brief, local application of caffeine that evokes concerted RyR opening, resulting in a robust, phasic Ca^2+^ transient and a subsequent contraction not dissimilar in shape to those evoked by electric stimulation.

Short caffeine pulses were also used by Su et al. [30] to compare the SR ability to take up Ca^2+^ in cardiomyocytes from different species. However, cells were internally perfused, which may affect cytosolic composition and cell volume, in contrast with the present experimental model, in which cell membrane is intact and the cytoplasmic environment is left undisturbed. Additionally, caffeine pulses were briefer (60 vs. 100 ms in [30]), which allows faster caffeine washout and less interference of prolonged Ca^2+^ release on the transient decay.

It was possible to confirm experimentally the important assumption that Ca^2+^ cycling was largely conservative in the FISRM, as the SR Ca^2+^ content was not significantly changed by a short series of successive caffeine-induced Ca^2+^ release events. This supports the notion that cytosolic Ca^2+^ removal can be attributable majorly to SERCA, with negligible contribution of the transporters not inhibited in this model (PMCA and MCU). These transporters seem to play a minor role in the Ca^2+^ transient decay in cardiomyocytes [5]. Also, changes in twitch Ca^2+^ transients and in SR Ca^2+^ content in mature cardiomyocytes were not observed after pharmacological PMCA inhibition [32]. Additionally, twitch Ca^2+^ transients under basal conditions are not affected by PMCA [33] or MCU knockout [34]. Simulation of Ca^2+^ transients with the mathematical FISRM closely reproduced the experimental results, which supports the assumption that the activity of only two Ca^2+^ transport paths (i.e., RyR and SERCA) is sufficient to account for the Ca^2+^ fluxes involved during the first caffeine-induced transients.

Nevertheless, one of the limitations of the experimental FISRM is that the amplitude of Ca^2+^ transients decays over repeated caffeine applications. Partial SR Ca^2+^ depletion due to cumulative removal of cytosolic Ca^2+^ by PMCA and MCU might be involved in this decay. The observation that a smaller progressive decline in the transient amplitude also in the mathematical FISRM, which included the Ca^2+^ flux via PMCA but not via MCU, lends support to this possibility, making it unlikely that changes in RyR sensitivity are involved. Nevertheless, results showed that the transient features and the SR Ca^2+^ load are largely preserved during the first five caffeine pulses, thus delimiting the reliable range of pulses.

SERCA stimulation and inhibition by ISO and tBQ, respectively, showed to result in opposite changes in twitch amplitude and in decline timecourse, as expected. ISO mimicks the β-adrenergic stimulation seen during sympathetic activation seen during exercise and the fight-or-flight (acute stress) response. In cardiomyocytes, β-adrenoceptor activation accelerates [Ca^2+^]i decay via SERCA stimulation and increases transient amplitude by augmenting L-type Ca^2+^ channel activity, increasing the SR Ca^2+^ content and, depending on the stimulation level, sensitizing RyR to Ca^2+^ (e.g., [27,31]. Enhancing both Ca^2+^ influx and the SR Ca^2+^ store contributes to increase the fraction of the SR Ca^2+^ load that is released at a twitch [6]. Simulation of ISO action was achieved by increasing both L-type Ca^2+^ channel permeability and SERCA affinity for Ca^2+^, resulting in accelerated [Ca^2+^]i decline in both types of contraction, but enhanced Δ[Ca^2+^]i only in electrically evoked twitches, in which greater Ca^2+^ current (absent in the FISR model) can trigger greater release of SR Ca^2+^ [6,35]. However, one might expect that increase in the fractional SR Ca^2+^ release (because of higher Ca^2+^ content) and/or RyR phosphorylation would lead to greater Ca^2+^ release and transient amplitude [6,8,33,36]. It is possible that the magnitude of these changes was insufficient to increase caffeine-induced Δ[Ca^2+^]i content release at this low, more physiological level of adrenergic stimulation.

As in the FISRM, Ca^2+^ influx via L-type channels is absent due to the removal of extracellular Ca^2+^, it is possible to isolate the effects of β-adrenergic stimulation on the SR function to better characterize, for instance, the role of RyR phosphorylation by different protein kinases at diverse sites in the modulation of systolic and diastolic SR Ca^2+^ release in intact cells from hearts under healthy and pathophysiological conditions [36,37,38], as well as in the search for therapies [26,39].

SERCA inhibition by tBQ, on the other hand, was simulated by diminishing the maximal Ca^2+^ uptake capacity, equivalent to a decrease in the number of active enzyme molecules, based on the observation that tBQ binds to the ATPase, forming a dead-end complex with interruption of the Ca^2+^ transport cycling [24,40]. In the experimental model, tBQ effects were on average stronger on amplitude of electrically stimulated twitches than of caffeine-induced transients, which was not reproduced by the simulations, in which a milder and comparable depression of Δ[Ca^2+^]i was observed in both kinds of transient. It is likely that these discrepancies are related to the inhibition of L-type Ca^2+^ current by this agent at concentrations required for SERCA inhibition [25], which was eliminated by extracellular Ca^2+^ removal and not included in the computational model as one of the mechanisms affected by tBQ.

Regarding the [Ca^2+^]i decline before and after treatment with ISO and tBQ, although able to simulate the qualitative effects of ISO and tBQ observed experimentally (i.e., acceleration and prolongation, respectively), the computational FISRM could not replicate the decay timecourse, showing a stronger effect in the late phases of the transient decay. This might be explained by the formulation of passive cytosolic Ca^2+^ buffers solely at equilibrium in the model of Bondarenko et al. [16], when the existence of fast buffers has been reported in cardiomyocytes [41]. Another possibility is the appearance of biphasic [Ca^2+^]i decline in the presence of agents that promote SR Ca^2+^ leak, such as ISO [42]. Curiously, tBQ has been reported to also transiently stimulate SR Ca^2+^ leak in skeletal muscle [43]. Thus, more studies should be performed to improve the models, and to explore the regulation of mechanisms involved in Ca^2+^ release.

In conclusion, simulations closely reproduced the experimental results, in agreement with the assumptions of the model. Thus, the proposed hybrid FISRM seems to represent a valuable tool for accessing and investigating the SR function in intact cardiac myocytes in pathophysiological conditions, and in the development and test of pharmacological therapies targeting the SR. Nevertheless, the use of this experimental model might also be extended to other cell types (e.g., skeletal and smooth muscle) to investigate Ca^2+^ mobilization from the sarco/endoplasmic reticulum by RyR, without the interference of transsarcolemmal Ca^2+^ fluxes.

## Figures and Tables

**Figure 1 bioengineering-12-00627-f001:**
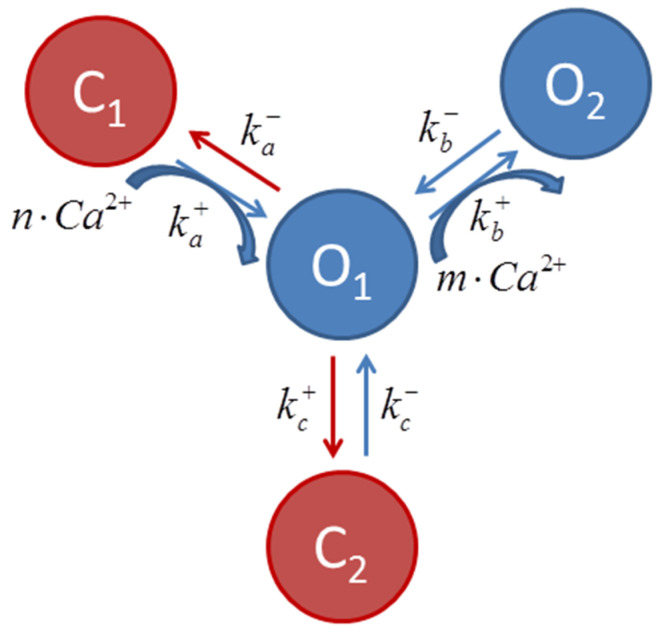
Scheme describing transition among ryanodine receptor states, where $k_{x}^{+}$ and $k_{x}^{-}$ are transition rates, and *m* and *n* are the cooperativity parameters of the Ca^2+^ binding determined experimentally [28]. The transitions C1 ↔ O1 ↔ O2 (curved blue arrows) are Ca^2+^ dependent. It is considered that most channels are in the C1 state at diastolic [Ca^2+^]i.

**Figure 2 bioengineering-12-00627-f002:**
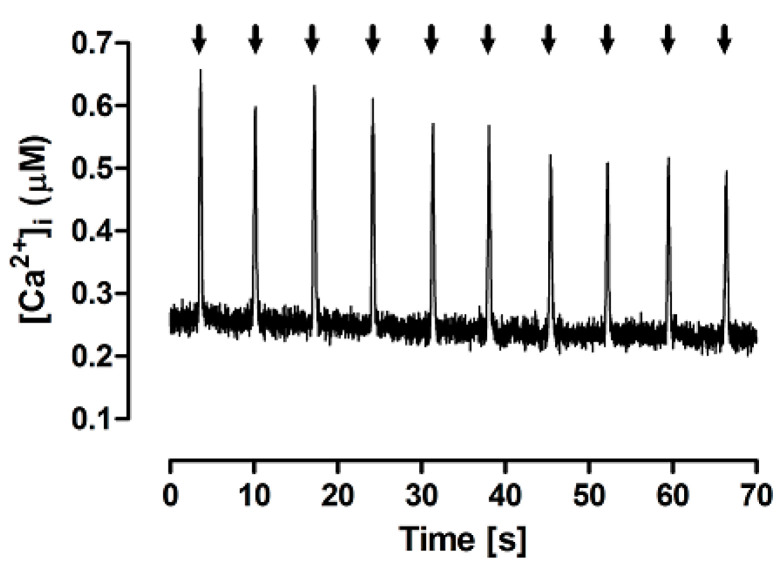
Typical Ca^2+^ transients evoked by brief (60 ms) caffeine pulses (arrows) in the experimental FISRM.

**Figure 3 bioengineering-12-00627-f003:**
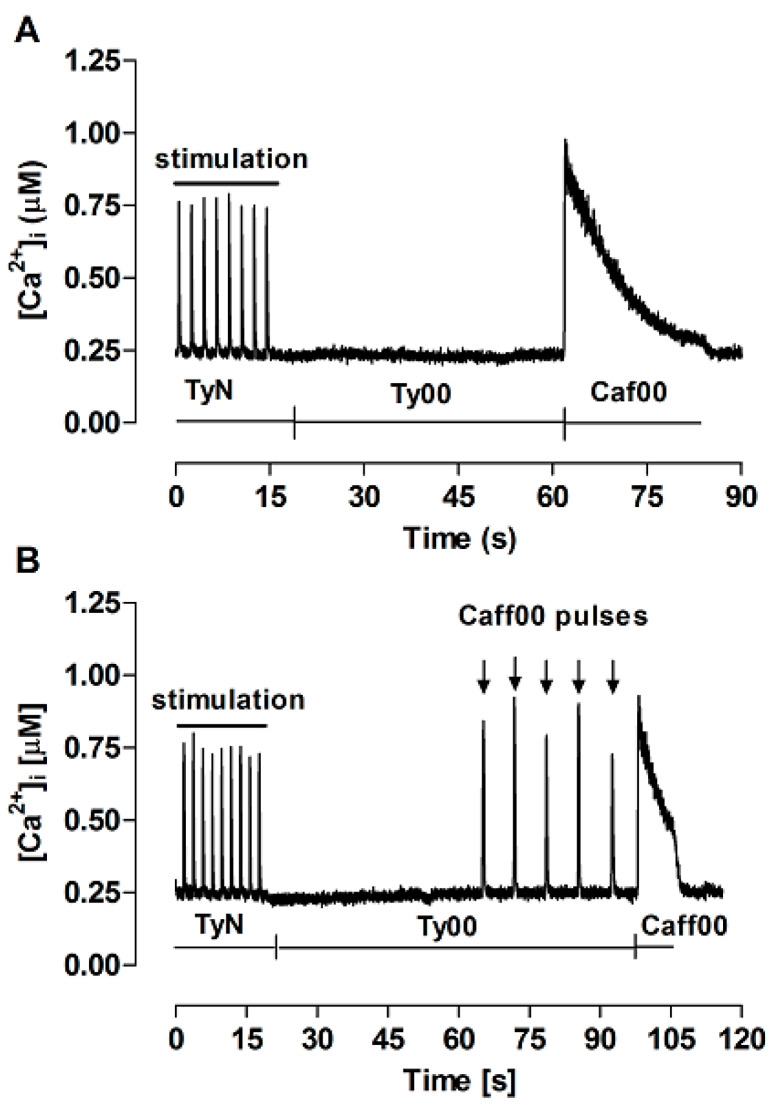
Ca^2+^ transients recorded for estimation of the SR Ca^2+^ content. TyN: perfusion with modified Tyrode’s solution, during which twitch transients were electrically evoked at 0.5 Hz; Ty00: perfusion with Ca^2+^, Na^+^-free Tyrode’s solution; Caff00: perfusion with Ty00 containing 10 mM caffeine sustained for at least 10 s, not preceded by (**A**) or following brief Caf00 pulses (downward arrows) (**B**).

**Figure 4 bioengineering-12-00627-f004:**
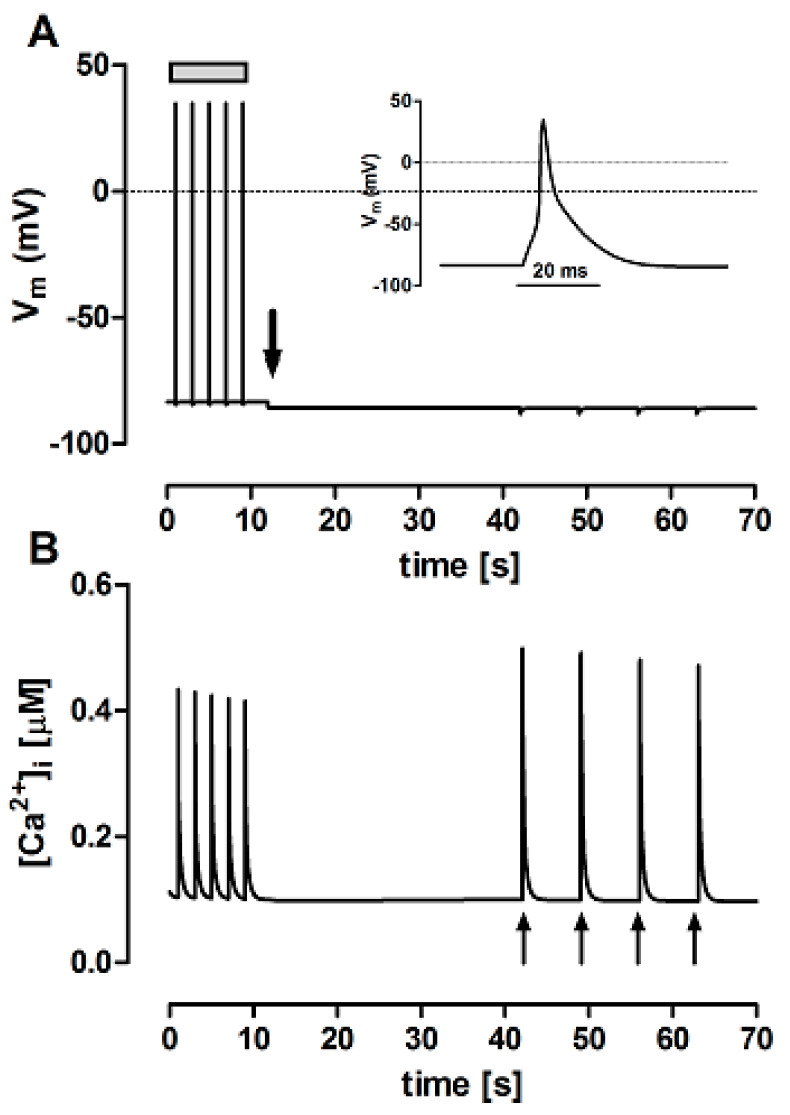
Simultaneous variations in membrane potential (V_m_, panel (**A**) and cytosolic Ca^2+^ concentration ([Ca^2+^]i, panel (**B**) simulated with the theoretical FISRM. Electrical stimulation (grey bar in (**A**) induced Ca^2+^ transients triggered by action potentials, one of which is shown on an expanded scale as the inset. After switching perfusion to Ty00 solution (downward arrow in (**A**)), electric diastole ensued, but Ca^2+^ transients could be evoked by application of caffeine pulses (upward arrows in (**B**)).

**Figure 5 bioengineering-12-00627-f005:**
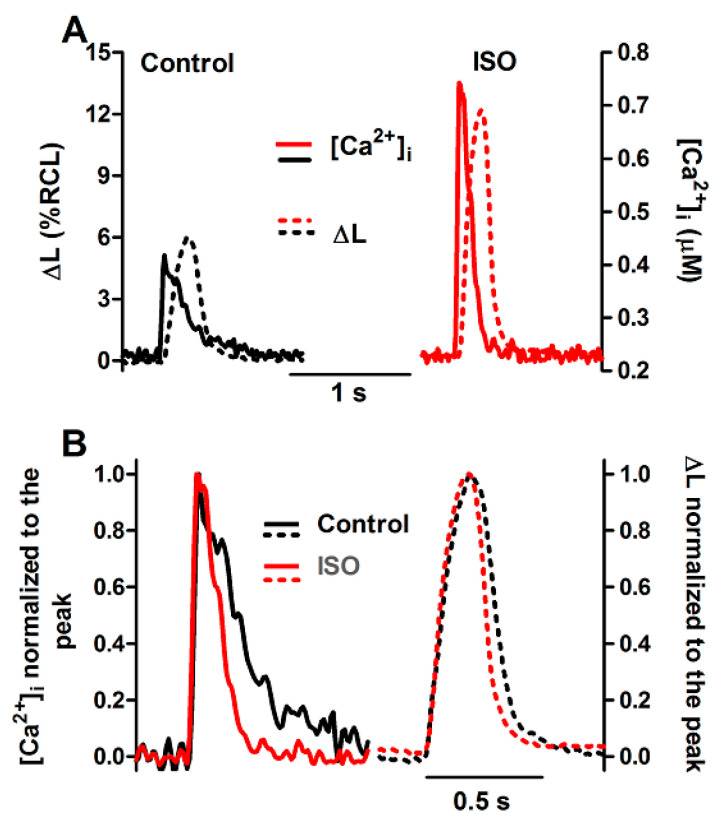
Ca^2+^ transients and twitch contractions recorded simultaneously at 0.5 Hz from a rat ventricular cardiomyocyte before (control) and after addition of isoproterenol (ISO, 10 nM). (**A**): Systolic cell shortening (ΔL, expressed as percent of resting cell length, RCL) and cytosolic Ca^2+^ concentration ([Ca^2+^]i). (**B**): the same traces after normalization to the respective peak value.

**Figure 6 bioengineering-12-00627-f006:**
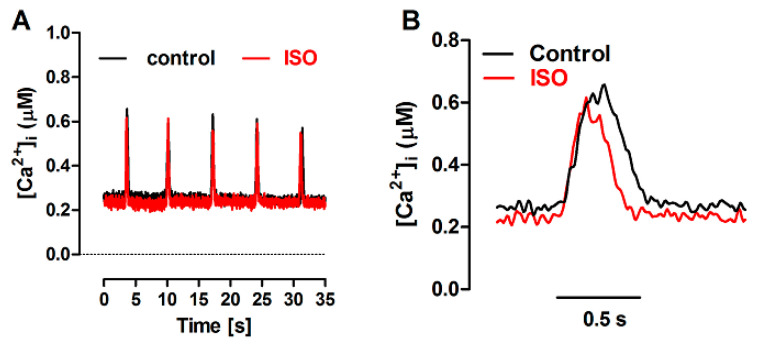
(**A**): Ca^2+^ transients evoked by caffeine pulses in the experimental FISRM before (control) and during exposure to 10 nM isoproterenol (ISO). (**B**): Single caffeine transients in an expanded timescale.

**Figure 7 bioengineering-12-00627-f007:**
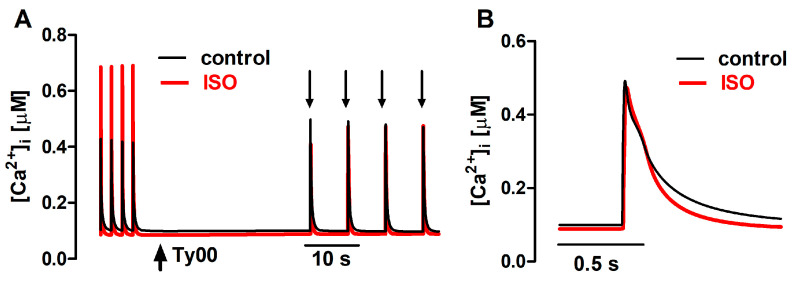
(**A**): Ca^2+^ transients generated with the mathematical FISRM, evoked by electrical stimulation and by caffeine pulses (downward arrows) in the absence and presence of isoproterenol (ISO). (**B**): Single caffeine-induced transients in an expanded timescale.

**Figure 8 bioengineering-12-00627-f008:**
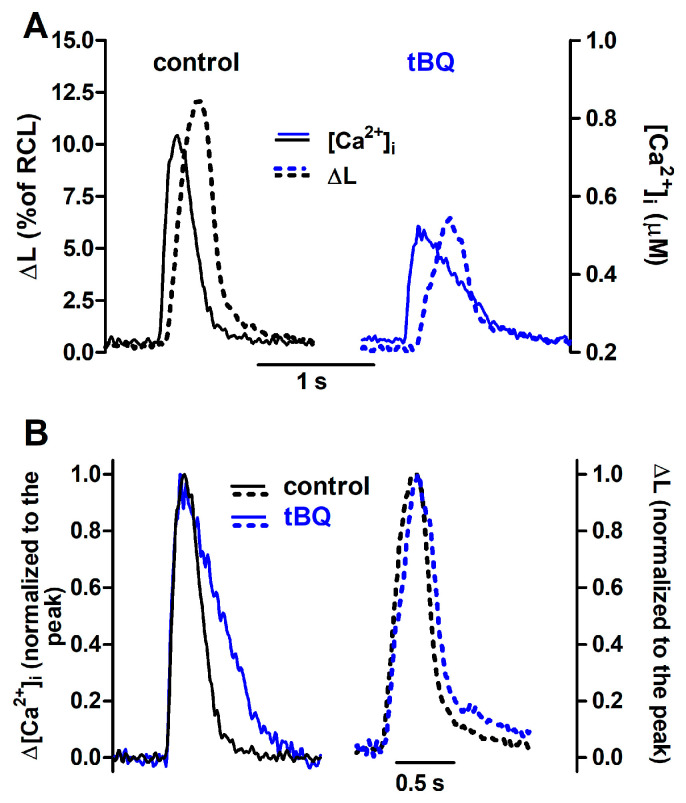
Ca^2+^ transients and twitch contractions recorded simultaneously from a rat ventricular cardiomyocyte before (control) and after addition of 2,5-di-(tert-butyl)-1,4-benzohydroquinone (tBQ, 5 μM). (**A**): Systolic cell shortening (ΔL, expressed as percent of resting cell length, RCL) and cytosolic Ca^2+^ concentration ([Ca^2+^]i). (**B**): the same traces after normalization to the respective peak value.

**Figure 9 bioengineering-12-00627-f009:**
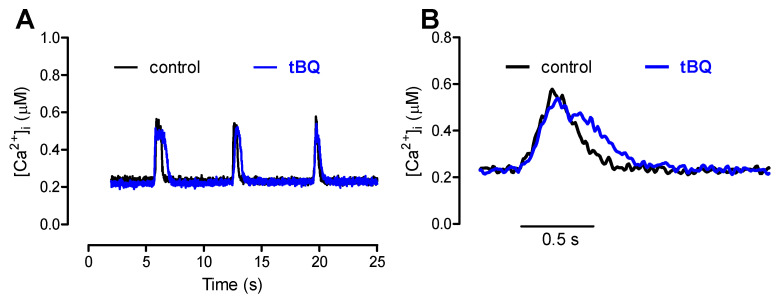
(**A**): Ca^2+^ transients evoked by brief caffeine pulses in the experimental FISRM before (control) and during exposure to 5 μM 2,5-di-(tert-butyl)-1,4-benzohydroquinone (tBQ). (**B**): Single caffeine-induced transients in an expanded timescale.

**Figure 10 bioengineering-12-00627-f010:**
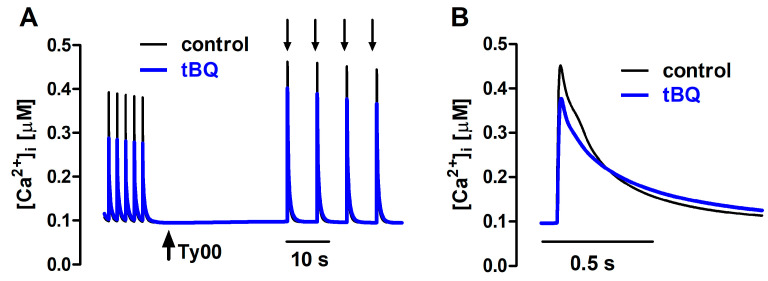
(**A**) Ca^2+^ transients generated with the mathematical FISRM before (control) and after exposure to 5 μM 2,5-di-(tert-butyl)-1,4-benzohydroquinone (tBQ), presented as described in Figure 7. (**B**) Single caffeine-induced transients in an expanded timescale.

**Table 1 bioengineering-12-00627-t001:** Effect of 10 nM isoproterenol (ISO) on the amplitude of electrically stimulated [Ca^2+^]i transients (Δ[Ca^2+^]i) and respective twitch contractions (ΔL, systolic cell shortening, expressed as percent of the resting cell length, RCL), on the decay half-time (t_0.5_) of both waveform types, and on the SR Ca^2+^ content ([Ca^2+^]_SR_, expressed as µM., i.e., µmoles Ca^2+^ per liter of non-mitochondrial cell water). Data are shown as the mean ± standard deviation of the mean (N = 7). *p* values: before (control) vs. after ISO, Student’s *t*-test for paired samples.

	Control	ISO	*p*
Δ[Ca^2+^]i (µM)	0.232 ± 0.083	0.440 ± 0.184	0.014
t_0.5_ [Ca^2+^]i (ms)	99 ± 16	74 ± 20	0.013
ΔL (% of RCL)	4.22 ± 2.30	8.10 ± 2.57	0.021
t_0.5_ ΔL (ms)	125 ± 37	96 ± 32	0.005
[Ca^2+^]_SR_ (µM)	95 ± 35	102 ± 36	0.007

**Table 2 bioengineering-12-00627-t002:** Effects of isoproterenol (ISO, 10 nM) and 2,5-di-(tert-butyl)-1,4-benzohydroquinone (tBQ, 5 µM) on the amplitude (Δ[Ca^2+^]i) and half-time of decline (t_0.5_ [Ca^2+^]i) of Ca^2+^ transients elicited by brief caffeine pulses. Data are shown as in Table 1.

	Control	ISO	*p*
Δ[Ca^2+^]i (µM)	0.324 ± 0.173	0.293 ± 0.183	0.471
t_0.5_ [Ca^2+^]i (ms)	125 ± 21	94 ± 16	<0.001
	**Control**	**tBQ**	** *p* **
Δ[Ca^2+^]i (µM)	0.272 ± 0.086	0.230 ± 0.056	0.029
t_0.5_ [Ca^2+^]i (ms)	117 ± 24	183 ± 43	<0.001

**Table 3 bioengineering-12-00627-t003:** Effect of 5 μM 2,5-di-(tert-butyl)-1,4-benzohydroquinone (tBQ) on the amplitude of electrically stimulated [Ca^2+^]i transients and respective twitch contractions. Data are presented as in Table 1.

	Control	tBQ	*p*
Δ[Ca^2+^]i (µM)	0.273 ± 0.170	0.199 ± 0.072	0.159
t_0.5_ [Ca^2+^]i (ms)	89 ± 12	155 ± 20	<0.001
ΔL (% of RCL)	7.06 ± 2.55	5.43 ± 1.92	0.049
t_0.5_ ΔL (ms)	97 ± 23	117 ± 24	<0.001
[Ca^2+^]_SR_ (µM)	112 ± 28	104 ± 29	0.005

## Data Availability

A link for the source-code of the mathematical FISRM is available in the Appendix A. The original contributions presented in this study are included in this article/Appendix A. Further inquiries can be directed to the corresponding author.

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
