# Peer review of "Functionally Isolated Sarcoplasmic Reticulum in Cardiomyocytes: Experimental and Mathematical Models"

_bioengineering, 2025, doi:10.3390/bioengineering12060627_

Round 1

Reviewer 1 Report

Comments and Suggestions for Authors

The paper entitled “Functionally isolated sarcoplasmic reticulum in cardiomyocytes: experimental and mathematical models” by Soriano, Bassani and Bassani, presents an in vitro biological model and its mathematical equivalent of the isolated sarcoplasmic reticulum, the structure that is essential in the excitation-contraction coupling in myocytes. The paper focuses on cardiomyocytes, but it might be valid for other myocyte types as well. As such, it might be helpful in studying causes and potential therapeutic solutions of muscle dysfunction in general; heart failure to begin with.

The paper is well written, the biological and mathematical models and the results of the experiments are clearly described; as such, I don’t have comments. The only suggestion I have is the addition of a Figure with the schematic representation of the mathematical model (like in the paper by Bodarenko).

Author Response

RESPONSE TO THE REVIEWERS

The authors appreciate the editorial decision and thank the reviewers for their positive feedback, comments, suggestions and criticism, which have contributed substantially to enhance the quality of the manuscript.

Please, find below the responses to the reviewers´ comments. The comments are shown in bold italic font, with their respective response shown below in blue. The changes made in the manuscript are indicated by the use of blue font.

We expect that the revised manuscript may be considered suitable for publication in Bioengineering.

 REVIEWER 1

COMMENT: The paper entitled “Functionally isolated sarcoplasmic reticulum in cardiomyocytes: experimental and mathematical models” by Soriano, Bassani and Bassani, presents an in vitro biological model and its mathematical equivalent of the isolated sarcoplasmic reticulum, the structure that is essential in the excitation-contraction coupling in myocytes. The paper focuses on cardiomyocytes, but it might be valid for other myocyte types as well. As such, it might be helpful in studying causes and potential therapeutic solutions of muscle dysfunction in general; heart failure to begin with.

The paper is well written, the biological and mathematical models and the results of the experiments are clearly described; as such, I don’t have comments. The only suggestion I have is the addition of a Figure with the schematic representation of the mathematical model (like in the paper by Bodarenko).

RESPONSE: We appreciate the reviewer´s suggestion. The additional figure with a diagram of the model (Fig. S1) was included in the R1 version of the supplement (see lines 224-225 of the article).

Reviewer 2 Report

Comments and Suggestions for Authors

In this manuscript, Soriano et al. introduce a novel hybrid approach termed the Functionally Isolated SR Model (FISRM) to investigate sarcoplasmic reticulum (SR) Ca²⁺ transport dynamics in intact cardiomyocytes. The authors effectively isolate SR Ca²⁺ fluxes from other cellular transport pathways by combining carefully designed experiments with a simplified mathematical model. Their method employs a Ca²⁺- and Na⁺-free extracellular medium along with a series of brief caffeine pulses to trigger SR Ca²⁺ release, enabling detailed analysis of Ca²⁺ uptake and release kinetics. The model was evaluated using compounds with opposing effects on SR Ca²⁺ uptake in rat ventricular myocytes. A theoretical simulation, adapted from previous models to fit the current experimental conditions, supported the main assumptions and reproduced key experimental findings. Overall, this is a well-executed and thoughtful study that presents a reasonable approach for investigating SR Ca²⁺ transport in cardiac cells. Thus, I recommend that the authors address the minor comments outlined below prior to publication.

  1. The authors used both "Ca²⁺" and "Ca2+" inconsistently throughout the manuscript. For consistency and clarity, it is recommended to use the proper scientific notation "Ca²⁺" consistently. Additionally, chemical formulas such as "CaCl2" and others should be correctly formatted with subscripts (i.e., CaCl₂).
  2. Can the authors include more data to show how well the model matches the experimental results?
  3. Did the authors confirm that other Ca²⁺ pathways were blocked in the Ca²⁺-, Na⁺-free medium, and were proper controls used?
  4. The authors are encouraged to clarify the reproducibility and stability of the caffeine-induced SR Ca²⁺ release protocol, especially regarding possible desensitization or depletion effects from repeated stimulation.
  5. Has the present model been tested in other species or cell types to see if it works more generally?
  6. In Figure 8B, the units for the x-axis (time in seconds) are missing and should be added.
  7. Several of the cited references appear to be outdated; the authors are encouraged to update the reference list by incorporating a few more recent and relevant studies that reflect current advances in the field.

Author Response

RESPONSE TO THE REVIEWERS

The authors appreciate the editorial decision and thank the reviewers for their positive feedback, comments, suggestions and criticism, which have contributed substantially to enhance the quality of the manuscript.

Please, find below the responses to the reviewers´ comments. The comments are shown in bold italic font, with their respective response shown below in blue. The changes made in the manuscript are indicated by the use of blue font.

We expect that the revised manuscript may be considered suitable for publication in Bioengineering.

REVIEWER 2

COMMENT: In this manuscript, Soriano et al. introduce a novel hybrid approach termed the Functionally Isolated SR Model (FISRM) to investigate sarcoplasmic reticulum (SR) Ca² transport dynamics in intact cardiomyocytes. The authors effectively isolate SR Ca² fluxes from other cellular transport pathways by combining carefully designed experiments with a simplified mathematical model. Their method employs a Ca²- and Na-free extracellular medium along with a series of brief caffeine pulses to trigger SR Ca² release, enabling detailed analysis of Ca² uptake and release kinetics. The model was evaluated using compounds with opposing effects on SR Ca² uptake in rat ventricular myocytes. A theoretical simulation, adapted from previous models to fit the current experimental conditions, supported the main assumptions and reproduced key experimental findings. Overall, this is a well-executed and thoughtful study that presents a reasonable approach for investigating SR Ca² transport in cardiac cells. Thus, I recommend that the authors address the minor comments outlined below prior to publication. 

  1. The authors used both "Ca²" and "Ca2+" inconsistently throughout the manuscript. For consistency and clarity, it is recommended to use the proper scientific notation "Ca²" consistently. Additionally, chemical formulas such as "CaCl2" and others should be correctly formatted with subscripts (i.e., CaCl).

RESPONSE: Thank you for this observation. Apparently, all subscript and superscript characters have lost their format at some point of file conversion. We have carefully revised the manuscript for correction of the unformatted characters.

  1. Can the authors include more data to show how well the model matches the experimental results?

RESPONSE: We are not sure if we have understood what kind of data the reviewer has in mind. We consider that one of the positive features of the FISRM is its simplicity, not only from the experimental aspect, but also from the mathematical point of view, as it is possible to eliminate the formulations related to most sarcolemmal ion currents and transport. So, considering the simplicity of the FISR system, testing the suitability of the mathematical model would be restricted to variation of only two kinds of Ca2+ flux between the SR and the cytosol: SR Ca2+ uptake and release. The former was pharmacologically stimulated and inhibited in the present report, with a reasonable match of the results. The Ca2+ release flux, on the other hand, should be explored in future studies, as it is more complicated to control experimentally. Nevertheless, we consider that, with exception of the temporal pattern of change in [Ca2+]i decline by the tested agents, the consistency between the models was satisfactory, especially the reproduction by the mathematical model of isoproterenol positive inotropic effect  for electrically stimulated transients, but not for those evoked by caffeine pulses (fig. 7 vs. data in tables 1 and 2). Another interesting coincidence between experiment and simulation results is the decrease in Ca2+ transient amplitude over successive caffeine pulses (please see response to comment 4).  

  1. Did the authors confirm that other Ca² pathways were blocked in the Ca²-, Na-free medium, and were proper controls used?

RESPONSE: This is very good point, indeed. Unfortunately, we were not able to block the pathways that remain operant in this medium, namely sarcolemmal Ca2+ ATPase (PMCA) and mitochondrial Ca2+ uniporter (MCU). Currently, pharmacological inhibitors of the former are not available for use in intact cells, as the classically used agents act on the intracellular face of the sarcolemma, and are not lipophilic and/or not selective for this ATPase. We were able to achieve successful PMCA inhibition in live cells using a cell-permeant form of carboxyeosin (doi: 10.1007/BF00373894; doi: 10.1152/ajpheart.00320.2001). However, this compound is no longer commercially available. On the other hand, inhibition of mitochondrial Ca2+ uptake can be rather tricky. The best results that we could achieve were by collapsing the electric potential across the internal mitochondrial membrane with a protonophore (cyanide), thus extinguishing the driving force for Ca2+ uptake. However, this also causes interruption of the ATP synthesis and can lead to cell deenergization and death. We could obtain functional recovery of the myocytes after ~1 h washout if cyanide exposure lasted < 1 min (doi: 10.1113/jphysiol.1992.sp019246; doi: 10.1152/ajpheart.00320.2001), but not longer. Unfortunately, such a short treatment would not be sufficient for running the caffeine pulse protocol, and even if it were, SERCA function would be impaired in an uncontrollable/predictable fashion. Thus, it was not possible to carry out the experiments under PMCA and MCU inhibition. However, as mentioned in the article, these transporters should not have important interference in SR Ca2+ content (at least for the initial caffeine-induced transients, please see response to comment 4), as they contribute <2% to cytosolic clearance during a typical Ca2+ transient (doi: 10.1113/jphysiol.1992.sp019246; doi:10.1113/jphysiol.1994.sp020130; doi: 10.1152/ajpheart.00320.2001). It is expected that progressive SR depletion occurs after many caffeine pulses, but we think that it should be relatively safe not to consider Ca2+ transport by the other pathways as long as the number of pulses is limited to 5, after which significant changes in SR Ca2+ content were not observed (manuscript lines 257-262). Additional information is provided in the R1 article on the impact of PMCA and MCU on Ca2+ transients (lines 394-398).          

  1. The authors are encouraged to clarify the reproducibility and stability of the caffeine-induced SR Ca² release protocol, especially regarding possible desensitization or depletion effects from repeated stimulation.

RESPONSE: SR Ca2+ partial depletion due to cumulative PMCA -dependent Ca2+ efflux and uptake into mitochondria may be a problem when successive caffeine-triggered transients are evoked without SR replenishment by Ca2+ influx. Changes in RyR function seem unlikely as a progressive decline in Ca2+ transient amplitude was observed also in the simulations, although the effect of caffeine on RyR sensitivity to Ca2+ remained unaltered. Moreover, the mathematical FISRM included PMCA, but not MCU function, which is in agreement with an increasing reduction in transient amplitude (lines 268-270), but smaller than that observed in the experimental model, in which MCU was functional (fig. 4 vs. fig. 2). Our previous results indicate that the Ca2+ taken up my mitochondria seems to eventually return to the SR in cells remain in the Na+,Ca2+-free medium, but this takes 3-5 min (doi: 10.1113/jphysiol.1993.sp019489), so that, within the timeframe of our protocol, the cumulative Ca2+ sequestration by this organelle, although small, might impact the SR Ca2+ content. For this reason, we have taken care to estimate this content after perfusion with 0Na+,0Ca2+ solution with and without the interposition of caffeine pulses (fig. 2). We observed that after 5 pulses the decrease in SR Ca2+ load was only < 5%, not attaining statistical significance, thus allowing the assumption that Ca2+ cycles within a conservative system, as in the mathematical FISRM.  For this reason, all following experiments were carried using only 4-5 caffeine pulses (lines 243-245). This limitation of the model was added to the Discussion section (lines 403-411).     

  1. Has the present model been tested in other species or cell types to see if it works more generally?

RESPONSE: So far, the model has been tested only in ventricular myocytes isolated from adult rats. Based on the observation that, during perfusion with 0Na+, 0Ca2+ solution, SR Ca2+ handling is rather similar in myocytes from rats, rabbits and ferrets (which show large differences in the interplay of Ca2+ transporters; e.g., doi: 10.1006/jmcc.1994.1152; doi: 10.1113/jphysiol.1992.sp019246; doi: 10.1113/jphysiol.1994.sp020131; doi: 10.1016/S0006-3495(95)80378-4), one might infer that the suitability of the model would not be species-dependent, at least among mammals. However, this premise awaits experimental confirmation.

  1. In Figure 8B, the units for the x-axis (time in seconds) are missing and should be added.

RESPONSE: Thank you for this observation. The figure was corrected accordingly.

  1. Several of the cited references appear to be outdated; the authors are encouraged to update the reference list by incorporating a few more recent and relevant studies that reflect current advances in the field.

RESPONSE: The large number of older references is explained by our preference to cite the original/classical papers rather than more recent reviews. We have certainly examined the recent literature, but, unfortunately, could not find much that was directly relevant to our manuscript. Nevertheless, 20% of the cited articles were published within the last 5 years, which we consider reasonable in view of the 40 year range covered by the bibliography.